# Innovative Therapeutic Approaches for Duchenne Muscular Dystrophy

**DOI:** 10.3390/jcm10040820

**Published:** 2021-02-17

**Authors:** Fernanda Fortunato, Rachele Rossi, Maria Sofia Falzarano, Alessandra Ferlini

**Affiliations:** Department of Medical Sciences, University of Ferrara, 44121 Ferrara, Italy; frtfnn@unife.it (F.F.); rssrhl@unife.it (R.R.); flzmsf@unife.it (M.S.F.)

**Keywords:** Duchenne muscular dystrophy, dystrophin restoration, antisense oligonucleotide chemistry, exon-skipping, stop codon reversion, gene therapy, innovative clinical trials

## Abstract

Duchenne muscular dystrophy (DMD) is the most common childhood muscular dystrophy affecting ~1:5000 live male births. Following the identification of pathogenic variations in the dystrophin gene in 1986, the underlining genotype/phenotype correlations emerged and the role of the dystrophin protein was elucidated in skeletal, smooth, and cardiac muscles, as well as in the brain. When the dystrophin protein is absent or quantitatively or qualitatively modified, the muscle cannot sustain the stress of repeated contractions. Dystrophin acts as a bridging and anchoring protein between the sarcomere and the sarcolemma, and its absence or reduction leads to severe muscle damage that eventually cannot be repaired, with its ultimate substitution by connective tissue and fat. The advances of an understanding of the molecular pathways affected in DMD have led to the development of many therapeutic strategies that tackle different aspects of disease etiopathogenesis, which have recently led to the first successful approved orphan drugs for this condition. The therapeutic advances in this field have progressed exponentially, with second-generation drugs now entering in clinical trials as gene therapy, potentially providing a further effective approach to the condition.

## 1. Introduction

The first descriptions of Duchenne muscular dystrophy (DMD) in the medical literature appeared in the first part of the 19th century with descriptions from Bell, as well as from Conte and Gioja. This was followed by a comprehensive account of the disease by Meryon in 1851, which recognized the X-linked recessive inheritance and the main muscle histopathological features following post-mortem analysis of affected boys. The disease was eventually attributed to Duchenne who, in 1868, examined a larger series of affected individuals, not only refining previous observations, but also introducing an innovative needle muscle biopsy technique that was used to assess the progressive nature of the muscle dystrophic pathology. Relatively little happened in the therapy field of DMD for more than a century until the identification of the *DMD* gene in 1987 [1]. The path to therapy development has been, however, slower and more convoluted than originally anticipated. This can be ascribed primarily to the complex role that dystrophin has in the muscles and to the rapidly progressive nature of the condition, with the loss of muscle tissue and its replacement by connective and adipose tissue, that complicate therapeutic attempts. The past few years, however, have seen the first successful attempts and the field rapidly evolve, providing benchmark knowledge for other neuromuscular disorders as well.

## 2. Molecular Genetics of DMD

Dystrophinopathies are due to mutations in the dystrophin (*DMD*, OMIM *30377) gene. Allelic heterogeneity is the rule, and all mutation types (large and small rearrangements like deletions and duplications, small mutations, splicing mutations, deep intronic mutations) occur in the *DMD* gene [2].

Despite this heterogeneity, about 75% of *DMD* mutations are copy number variations (CNVs), rapidly identifiable by hybridization-based procedures such as the multiple ligation probe assay (MLPA) and comparative genomic hybridization (CGH). Using these methods, the vast majority of DMD patients can be rapidly genetically diagnosed, with the finding of in-/out-of-frame deletions or duplications. According to the frame rule, in-frame deletions give rise to a milder dystrophinopathy variant known as Becker muscular dystrophy (BMD). The recent introduction of next-generation sequencing (NGS) technologies, mainly based either on single-gene or on gene panel testing, has greatly improved the accurate identification of small mutations [3,4], revealing a remarkable number of these variants in DMD and BMD patients. A very minor percentage of mutations, below 1%, is atypical and requires alternative methods, such as RNA analysis, to be identified. It is very likely that whole-genome sequencing (WGS), when optimized for accurate CNV detection and standardized as a diagnostic procedure, will represent the elective diagnostic strategy, able to detect all *DMD* mutation types at the DNA level [5].

Genetic diagnosis is mandatory in DMD and BMD patients for many reasons: (1) genetic counseling and family planning, including female carrier identification and pre-implantation or prenatal diagnosis; (2) genotype-phenotype correlation and frame rule implications, which may predict mild or severe phenotypes and therefore are relevant for delineating the natural history of the disease; and (3) personalized gene or mutation-specific therapies. 

Due to the availability of genetic counseling and prevention in families with affected boys, dystrophinopathies are currently often due to de novo mutations, since familial cases have become rare. This fact, together with the new therapeutic options, raises a number of issues related to early detection of patients [6], identification of carriers [7], and/or prenatal testing via non-invasive procedures such as those based on genetic testing of free fetal DNA circulating in maternal blood [8].

All of these aspects, together with the increased availability of targeted therapies, will be challenging for DMD.

## 3. Pathophysiology of DMD 

The *DMD* gene is transcribed into a 14 kb mRNA which gives rise to the translation of a 427 kDa protein. Three full-length isoforms are synthesized, driven from different promoters (the Dp427m, where “m” stands for muscle; the Dp427c, where “c” stands for the cortical brain promoter; and Dp427p, for the cerebellar Purkinje cell promoter, which are expressed prominently in muscle and heart, the brain and the cerebellum, respectively). In addition, multiple 3′ isoforms are transcribed from other internal gene promoters (Figure 1) [9,10,11]. These shorter isoforms are relevant for the function of dystrophin in the brain, peripheral nerves, and the retina [2]. These multiple isoforms and their different localization clearly point to multiple and distinct functions in humans. In skeletal and cardiac muscles, dystrophin has a predominant mechanical role, but it is also involved in cell signaling, mainly via the alpha-syntrophin and neuronal nitric oxide synthase (nNOS) circuit [12]. In the brain, the role of the major dystrophin isoforms is predominantly that of signaling and synaptic circuits modulation; however, some of the shorter isoforms are also involved in developmental aspects and in regulation of extracellular ion homeostasis [13].

### 3.1. Striated Muscle

In skeletal muscle, the dystrophin protein plays a key role in maintaining the integrity, flexibility, and stability of the sarcolemma by anchoring the intracellular actin cytoskeleton to the extracellular matrix through the dystrophin-glycoprotein complex (DGC). The deficiency of dystrophin leads to sarcolemma damage by contractile forces, especially eccentric exercise, resulting in increased permeability of ions and small molecules [14]. Indeed, Ca^2+^ homeostasis is dysregulated in dystrophic muscle since increased entry of calcium ions in muscle fibers that activate protein degradation and higher levels of reactive oxygen species (ROS) has been observed [15]. This determines continuous cycles of muscle degeneration and regeneration that eventually cannot be compensated by the regenerative capacity of the satellite cells, resulting in the replacement of muscle by fibrotic, connective tissue and fat [16].

Intriguingly, dystrophin is also transiently produced in the satellite cells (quiescent muscle stem cells located under the basal lamina of muscle fiber), where it is responsible for controlling their polarity and asymmetric division. Therefore, a lack of dystrophin in satellite cells affects symmetrical division, reducing the generation of new myogenic progenitors and impairing muscle regeneration [17]. This indicates that the disease progression in DMD is not due to muscle stem cell depletion but is caused by intrinsic satellite cell dysfunction.

Interestingly, the disturbance of asymmetric division of stem cells is associated with the growth and progression of cancer [18]. It was recently reported that dystrophin is a tumor suppressor gene and likely an anti-metastatic factor in myogenic sarcoma and some brain tumors, suggesting that therapeutic approaches developed for muscular dystrophies may also serve in the treatment of cancer [19,20].

Regarding the role of dystrophin in signaling, this is mediated via the interaction with proteins such as actin, β-dystroglycan, syntrophins, and α-dystrobrevin which, in turn, are linked to voltage-gated Na^+^ channels, nNOS, and stress-activated protein kinase-3 (SAPK3) [12]. Dystrophin and syntrophin proteins interact specifically with the PDZ domain-binding motif of the cardiac voltage-gated sodium channel. Dystrophin deficiency alters these interactions and impairs the expression and localization of functional Na^+^ channels in the cardiomyocyte membrane [21].

In addition to the heart, the lack of dystrophin affects the voltage-dependent L-calcium channels, one of the principal channels involved in excitation-contraction coupling in the cardiac muscle. These channels are linked to F-actin by subsarcolemmal proteins, including dystrophin [22]. Disruption of actin filaments enhances the calcium influx through L-calcium channels and may contribute to cardiomyopathy development [23].

The mislocalization of nNOS, in the absence of dystrophin at the sarcolemma, results in a secondary disruption of muscle nitric oxide (NO) signaling, which leads to abnormal regulation of blood flow within exercising skeletal muscle, causing a paradoxical vasoconstrictor response, which is believed to exacerbate the pathology. In particular, the impaired NO production in dystrophin-deficient *mdx* mice is due to a disrupted mechanotransduction AMP-activated protein kinase (AMPK)-nNOS signaling, suggesting that AMPK may be a possible pharmacologic target to restore NO synthesis in DMD [24,25].

### 3.2. Brain

The deficiency of dystrophin in the brain is directly correlated to the complex neuropsychiatric phenotype that affects up to 50% of DMD boys and a smaller proportion of BMD patients, encompassing intellectual disability, autism, and attention deficit disorder [26,27]. 

As shown in Figure 1, the brain expresses the full-length Dp427 and the shorter Dp140 and Dp71 isoforms [2], but the exact distribution of each specific isoform along the human brain areas, in terms of RNA and protein expression, is not yet well defined. The ongoing Brain Involvement iN Dystrophinopathies (BIND) project aims at addressing this crucial aspect and will elucidate the role of dystrophin in the brain [28].

The available information concerning the role of isoforms and their expression in the brain is limited. It has been reported that full-length Dp427 isoforms are highly expressed in the hippocampus and amygdala of the adult human brain, while low expression levels are found in the cerebellum [29]. Dp427 isoforms have a role in anchoring and clustering gamma-aminobutyric acid type A (GABAA) receptors at post-synaptic membranes, where it regulates the GABAergic synaptic function and glutamatergic synaptic plasticity [30]. 

Regarding the short dystrophin isoform, a recent study identified the expression of Dp140 not only during the developmental stages but also in the adult brain, with the highest expression in the cerebellum [29]. 

The ubiquitous Dp71 isoform also has a role in glutamergic neurotransmission, and in addition, it influences developmental myelination and extracellular ion regulation [31].

The severity of central nervous system (CNS) co-morbidities is directly related to the site of the mutation in the *DMD* gene and, in turn, on how many isoforms are deficient in the CNS of affected DMD individuals [27,32,33].

## 4. Rationale for Current Therapeutic Interventions in DMD 

Two main approaches are being pursued for DMD: (1) the restoration of dystrophin (or of dystrophin surrogate molecules) at the sarcolemma in order to improve the structural integrity of muscle fibers or (2) therapeutic attempts dealing with secondary consequences of dystrophin deficiency and the progressive dystrophic pathology. 

Regarding the first approach, there are multiple strategies under investigation. Some take advantage of splice-switching antisense oligonucleotides (AONs) that can be used in patients with specific out-of-frame deletions that can be trimmed during the pre-mRNA splicing to a shorter but in-frame message. This tries to recapitulate the mutation effect occurring in BMD individuals in which a qualitatively and quantitatively different dystrophin protein can be produced. An alternative approach is the replacement of the *DMD* gene using viral vectors (typically adeno-associated virus (AAV) gene therapy) containing and delivering mini- or micro-dystrophin constructs. Other therapeutic approaches are also focused on the pharmacological upregulation of other proteins physiologically present in muscle, such as utrophin, which have similar properties as dystrophin [34].

Regarding the attempts of dealing with secondary consequences, these strategies target different aspects of disease pathology, ranging from inflammation to fibrosis to muscle mass and regeneration, to muscle blood flow.

### 4.1. Restoring Dystrophin Protein Production

The intuition that RNA manipulation via splicing modification could be applied to DMD to correct mutations first appeared in the literature in the 1990s [35,36]. Matsuo [37,38] described a mutation in the so-called DMD Kobe patient by inducing exon 19 skipping, meaning exon omission from the spliced *DMD* mRNA induced an out-of-frame deletion. Other similar observations followed, and Nicholson [39] and Sherrat [40] independently raised the concept of dystrophin rescue due to internal spontaneous exon skipping in DMD boys, providing, for the first time, an explanation of the revertant fiber event and introducing the concept of dystrophin protein restoration, today so popular in clinical trials [34,41,42]. Indeed, the dystrophin protein internal region, the rod-shaped domain formed by 24 spectrin-like repeats, seems to be tolerant to in-frame deletions that do not severely compromise its function, as clearly suggested by patients with BMD [43,44].

The therapeutic use of exon skipping was first described in 1996 [36]. This pioneering paper introduced the novel approach that aimed at artificially modulating the exon incorporation into mature RNA via short, anti-oriented molecules (antisense oligoribonucleotides (AONs)) which, by complementary hybridizing exonic splicing enhancer sequences (ESE), can mask the exon per se and cause its omission from the messenger RNA (Figure 2).

The principle of exon skipping encountered immediate success, being well applicable to all deletion mutations that can be corrected by inducing favorable exon skipping, able to reframe the transcript and to rescue the dystrophin protein translation, generating a BMD-like molecule [45].

When designing AON-mediated therapies, two issues were identified as outstanding: (1) specificity, meaning reaching the correct exon without off-targeting effects and (2) stability, which protects the fragile and labile RNA molecule from RNAase degradation and makes it able to increase its half-life and get on target.

All AONs currently in clinical trials or designated as orphan drugs are chemically modified and strengthened via backbones. Table 1 summarizes the most recent clinical trials based on the AON-mediated exon-skipping strategy.

The most used AON chemistries are the phosphorodiamidate morpholino oligomers (PMOs) and the 2′-O-methyl-phosphorothioates (2′OMePS). Table 2 summarizes the chemistry and features of the main AONs used for DMD therapy.

PMO AONs have a modification in the sugar where a morpholino ring replaces the furanose and a phosphorodiamidate linkage is formed between the nitrogen at the morpholino ring and the hydroxyl group at the 3′ residue. PMOs were first developed by Antivirals, later renamed AVI-BioPharma, and more recently by Sarepta Therapeutics. PMO AONs are very stable, neutrally charged, and highly hydrophilic. This confers an excellent safety profile to PMO molecules that cause low off-target effects and immune responses [46]. Based on these promising features, several compounds were developed, such as eteplirsen (Exondys 51; NCT03218995, NCT03985878; NCT03992430; NCT04179409), designed to skip exon 51 and effective in DMD patients with amenable deletions, ~15% of deleted DMDs. Eteplirsen was recently approved (conditional approval to be confirmed in 2021) as an orphan drug by the US Food and Drug Administration (FDA), while in the European Medicines Agency (EMA), its evaluation is underway. 

Other PMOs for skipping exon 45 (casimersen) and exon 53 (golodirsen and viltolarsen) are now in randomized placebo-controlled clinical trials (NCT02500381; NCT04179409; NCT03532542; NCT04060199; NCT04337112). 

Golodirsen received its first approval on 12 December 2019 in the United States. The FDA gave conditional approval for casimersen and viltolarsen in 2020 and their final approval will be discussed following the results of the ongoing phase 3 ESSENCE (NCT03532542) and RACER53 (NCT04060199) studies.

So far, PMO AONs appear to be well-tolerated and safe in DMD patients following weekly intravenous (IV) administration [47,48,49]. However, the neutral charge of PMOs represents a limitation in AON cellular uptake and gives fast clearance in the bloodstream, therefore reducing the therapeutic effect [50]. Recent data suggest a role of macrophages as local drug reservoirs. Macrophages are recruited in inflammatory foci associated with dystrophic lesions, and incorporating PMO molecules, they can maintain PMO availability in myogenic cells over a long time. In addition, myogenic precursors seem to vehicle the direct delivery of PMOs into regenerating myofibers [51].

Next-generation chemistry of PMOs conjugated with cell-penetrating peptides (peptide phosphorodiamidate morpholino oligomers (PPMOs)) was developed to increase cell penetration, exon skipping, and dystrophin levels. Significantly improved muscle targeting and dystrophin rescue were observed in the *mdx* animal model and also in cardiac muscle targeting, which is otherwise not sufficiently targeted by naked PMOs [52].

While the safety profile of first generation PPMOs raised some concerns after preclinical studies, the new generation of PPMOs that use Sarepta’s PPMO chemistry (SRP-5051) have recently entered a phase I/II clinical trial (NCT03675126; MOMENTUM, NCT04004065). 

Sarepta Therapeutics announced positive results from part A of the MOMENTUM study. They observed consistently higher tissue exposure, exon skipping, and dystrophin production in patients taking a monthly dose of SRP-5051. In addition, SRP-5051 was found to be well tolerated across all doses, supporting next-dose escalation studies and further clinical development [53].

2′OMePS are AONs with a phosphorothioate (PS) backbone and 2′-O-modification of the ribose residue. These chemical modifications result in higher efficacy, nuclease resistance, and increased bioavailability [54]. Conversely, PS modification causes toxicity and adverse effects due to the affinity of phosphorothioates to plasma proteins, which consequently prevent their urinary excretion and promote their retention in other organs such as the kidneys and liver [55]. One early 2’OMePS in clinical development, drisapersen (developed by Prosensa Therapeutics, GlaxoSmithKline, and Biomarin), was abandoned after several phase II and III studies, mainly due to severe adverse events, which included injection side reactions, proteinuria, and thrombocytopenia [56,57,58]; (NCT01890798; NCT01910649; NCT02636686). The FDA did not give regulatory approval, and the application to the EMA was withdrawn and the study terminated [59,60].

A recent development has been the synthesis of the stereopure AON. When AONs are synthesized, they consist of a mixture of stereoisomeric molecules, which may cause off-target effects [56,61]. Obtaining stereochemically pure drugs, with a reduced component of stereoisomers, may improve the safety profile and the efficacy. Recently, a scalable synthetic process, able to produce stereopure 2′OMePS AONs, was set up [62]. Wave Life Sciences developed suvodirsen (previously known as WVE-210201), an investigational stereopure oligonucleotide able to skip exon 51. In a phase 1 trial (NCT03508947), suvodirsen resulted as safe and well tolerated, but following the analysis of dystrophin levels in the muscle from DMD patients after 12 or 22 weeks of treatment, significant dystrophin restoration was not observed and the study was terminated [63].

### 4.2. Ataluren as a Read-Through Strategy for Nonsense Mutations

A different mutation-specific approach was developed by PTC Therapeutics with the development of ataluren, an orally bioavailable drug that targets the nonsense mutations found in approximately 10–15% of DMD patients. The read-through of nonsense mutations is expected to lead to the ability of producing some dystrophin protein. Despite two randomized, placebo-controlled studies (NCT00592553 and NCT01826487) showing a slowing of disease progression provided by ataluren, no significant effect on dystrophin restoration was detected. The reasons can be found in the patients’ loss of target tissues and in the relatively short clinical trial’s time [64]. The authors tested ataluren efficacy by collecting trial data to explore the intent-to-treat (ITT) populations and two patient baseline 6-min-walk-distance (6MWD) subgroups. From a clinical point of view, the subgroups included patients in the ambulatory transition and ambulatory decline disease phases. The drug’s effect was more significantly detected in the subgroup of patients (pre-specified in the second study, NCT01826487) who were in the intermediate stages of ambulatory decline, i.e., with a 6MWD between 300 and 400 m. Importantly, the lack of efficacy in the better-functioning patients was considered to be related to their stability during a 48-week clinical trial, which is the result of the improved standards of care and of the widespread use of corticosteroids that play an unequivocal role in slowing down disease progression. Ataluren was well tolerated, and this contributed to its conditional approval by the EMA, while the drug is being considered by the FDA [65].

### 4.3. AAV Gene Therapy

Gene therapy is a promising approach that uses artificial genes to replace a defective gene or to modify its sequence or its expression with the aim of potentially treating all genetic diseases [66]. Most of the current gene therapy treatments require the use of viral vectors to deliver the artificial gene constructs. 

AAVs are the privileged gene vectors used for the development of gene therapy approaches because of their multiple-tissue-targeting ability, such as the retina, central nervous system, liver, and muscle [67]. The high infection efficiency and the large encapsidation capacity make AAVs the most appealing vectors for delivering modified genes [66].

Several AAV serotypes that can be used for gene therapy, having also different tissue affinity, for example, serotypes 1, 6, 8, and 9, exhibit a potent tropism for striated muscles, whereas AAV9 has excellent cardiac tropism [68].

Four main categories of AAV-delivered gene therapy (gene replacement, modifier gene expression, gene editing, and gene lockdown) are emerging as a potential treatment for several types of muscular dystrophies [69].

Gene replacement is an ideal method for treating monogenic, autosomal, or X-linked recessive diseases like DMD or beta-thalassemia, where the gene of interest can fit into the delivery vector [70,71].

A limitation of AAVs, however, is the carrying capacity (∼5 kb) available for artificial genes and regulatory cassettes (RCs). These size constraints are problematic for the size of the *DMD* gene (the full-length transcript being 14 kb), well over the carrying capacity of AAVs. Due to the difficulty in virus packaging with the full-length *DMD* gene, a truncated form of dystrophin (mini- and micro-dystrophin) was developed [72].

Partially functional μDys have been shown to protect the sarcolemma from contraction-induced injury and increase force generation, thus improving the dystrophic pathology in striated muscles of dystrophic mouse and canine models for DMD [73,74]. However, dosage-dependent immune responses triggered by AAV-mediated gene therapy, especially for high-dose delivery, are the major safety and tolerability concerns that limit its clinical application. An inflammatory myopathy was described in two dogs treated with a high dose (1.5 × 10^14^ vector genomes (vg)/kg) of AAV administered via intravascular injection [75]. Another study reported transient induction of pro-inflammatory cytokines after intravascular delivery of AAV2 and AAV8 capsids and showed that stronger adaptive immune responses correlate with higher delivery dosages [76]. Presumably, the induction of immune responses against mini-dystrophin was also the cause of the failure of the first mini-dystrophin clinical trial. Treated patients failed to express the synthetic mini-dystrophin; on the contrary, the specific T cell cluster against mini-dystrophin was detected even when the protein was not expressed [77].

A recent option is the development of a miniaturized dystrophin-related protein, Utrophin (µUtro), which is highly functional and non-immunogenic. The authors found that histological and biochemical markers of myonecrosis and regeneration were completely suppressed following systemic administration of AAV-μUtro to neonatal dystrophin-deficient *mdx* mice. More importantly, no evidence of a cell-mediated immune response against µUtro was detected, suggesting utrophin-derived therapies as an option in treating clinical dystrophin deficiency [78].

Three clinical trials utilizing micro-dystrophins are ongoing in the United States (Table 3). A Pfizer phase 1 study is investigating dose, safety, and tolerability of a single IV infusion of PF-06939926, an AAV9-mediated transfer of micro-dystrophin, in ambulatory and non-ambulatory DMD patients (NCT03362502). Another open-label phase 1/2 trial, sponsored by Sarepta Therapeutics (NCT03375164), is examining the safety of IV infusion of rAAVrh74.MHCK7 micro-dystrophin. The AAV9-vector-based micro-dystrophin transfer through SGT-001 was investigated by Solid Bioscience, but the clinical trial was suspended due to safety concerns (NCT03368742) [79].

Sarepta’s interim data appeared encouraging: a higher initial dose of 2 × 10^14^ vg/kg appears safe, tolerated, and associated with a higher efficacy. At 12 weeks, muscle dystrophin levels demonstrated a mean of 81.2% muscle fibers expressing micro-dystrophin with a mean intensity at the sarcolemma by immunohistochemistry of 96% compared to normal biopsies. All treated patients showed functional improvement on the North Star Ambulatory Assessment (NSAA) scale and reduced creatine kinase (CK) levels that were maintained through one year [80].

An alternative strategy to target a DMD/muscle damage-related pathway was also proposed by Kevin Flanigan in a recent ongoing clinical trial, utilizing AAV delivery of beta-1,4 N-acetylgalactosaminyltransferase 2 (*GALGT2*), a GalNac glycosyltransferase (NCT03333590). Because it does not replace the defective gene itself (the *DMD* gene), the therapy is called surrogate gene therapy. *GALGT2* overexpression has been shown to inhibit the development of the disease in different forms of muscular dystrophy: in the *mdx* model of DMD, congenital muscular dystrophy 1A, and limb girdle muscular dystrophy 2D. In these cases, the improvement of disease phenotypes was related to GALGT2 overexpression, inducing the glycosylation of α-dystroglycan and the upregulation of dystrophin and laminin α2 surrogates [81].

A number of experiments have demonstrated that *GALGT2* overexpression can prevent injury in *mdx* and wild type (WT) skeletal muscle. In particular, *GALGT2* overexpression causes a significant reduction in force drop during eccentric contractions and could improve the maximal specific force, conferring increased resistance to muscle damage, even in the absence of dystrophin [82,83]. Recently, it was also shown that altering the glycosylation of the cardiomyocyte sarcolemma membrane by overexpression of *GALGT2* can prevent the loss of cardiac function in the *mdx* mouse heart [84]. All these data have encouraged the development of a pre-clinical and now also clinical program to use rAAVrh74.MCK.GALGT2 to treat patients with DMD and other forms of muscular dystrophy.

AAV-mediated mini-/micro-dystrophin transfer appears to be one of the most promising therapeutic approaches, but several challenges (i.e., transfection efficiency, administration method, and immune response) are still present and need to be carefully assessed.

### 4.4. Future Perspectives

#### 4.4.1. CRISPR/Cas9 Mediated Gene Editing 

CRISPR/Cas systems are popular genome-editing technologies that belong to a class of programmable nucleases. Due to their simplicity, speed, and efficiency in modifying endogenous genes in any cell or target tissue, they have revolutionized basic science research.

These nucleases produce specific changes in regions of interest in the genome by inducing targeted double-strand breaks (DSBs) on chromosomes, followed by reparation through cellular mechanisms. Repair mechanisms include non-homologous end joining (NHEJ), which is prone to error, and homology-directed repair (HDR), which is error-free. They permit the generation of mutations (e.g., insertions, deletions, or substitutions in the targeted area) that may interrupt, eliminate, or correct the gene defects [85,86]. The programmable nucleases comprise meganucleases [87], zinc-finger nucleases [88], transcription activator-like effector nucleases [89], and the CRISPR/Cas9 system [90]. In detail, CRISPR/Cas9 genome editing is based on a guide RNA (gRNA) that directs the Cas9 endonuclease to create a DSB in a specific target area. Subsequent DNA repair results in re-ligation of the broken ends and genomic modifications at the target site, thus allowing rapid, easy, and efficient modification of endogenous genes in various cell types [91]. Genetic diseases most amenable for CRISPR-Cas9 editing are those in which a single allele needs to be targeted, as biallelic targeting is associated with much lower efficiency [92].

In the DMD context, recent studies have revealed that the delivery of CRISPR genome-editing tools by AAVs can reframe the mutated *DMD* gene and restore dystrophin expression in both DMD patient cells and in short-term mouse studies in vivo [93,94]. A widely used model is *mdx* mice carrying a nonsense mutation in exon 23 of the *DMD* gene that disrupts the transcription of *DMD* mRNA and the expression of the DYS protein [95]. This mutation can be removed by co-delivery of an AAV-Cas9 vector that expresses Staphylococcus aureus Cas9 and an AAV-gRNA vector leading Cas9 to introns 22 and 23 [96].

Several independent groups recently reported that this approach successfully resulted in excision of the mutation, thus enhancing the expression of truncated but functional dystrophin in myofibers, cardiomyocytes, and muscle stem cells (MuSCs) [93,96,97,98].

CRISPR/Cas9 editing is designed, in most cases, to restore the *DMD* open reading frame (ORF) and to generate a truncated but partially functional dystrophin protein through simulating an AON-skipping effect but acting at the genomic level.

An efficient correction strategy of exon 44 deletion mutations by CRISPR-Cas9 gene editing was demonstrated in cardiomyocytes obtained from patient-derived induced pluripotent stem cells (iPSCs) and in a mouse model harboring the same deletion [99]. Similarly, the DMD phenotype of a mouse model deleting exon 50 was corrected by systemic delivery of recombinant AAV9-packaged Cas9 nuclease and single-guide RNAs (sgRNAs) [94].

The efficacy and safety of this approach were recently confirmed in large mammals by the same authors. Their results revealed the efficacy of single-cut genome editing for restoring dystrophin expression in a deltaE50-Md canine model of DMD, reaching up to ~80% of WT levels in some muscles after 8 weeks of treatment. An improved muscle histology was also observed in treated dogs [100].

A limitation to this approach might be represented by mutations that cannot be corrected by genome editing as large deletions of essential regions of the N- or C-terminal domains [101]. Despite promising results from these short-term studies, clarification is needed concerning whether one-time systemic AAV CRISPR therapy can lead to persistent and life-long mutation correction, considering the chronic nature of DMD.

This critical question was addressed by treating a 6-week-old *mdx* mouse with a single IV injection of AAV-9 CRISPR therapy and evaluating long-term dystrophin expression and disease rescue at 18 months. The authors confirmed and extended findings of short-term studies: through a modification of CRISPR-editing machinery (i.e., an increase of the gRNA vector dose), they demonstrated for the first time that systemic AAV CRISPR editing can lead to dystrophin restoration and reduction of fibrosis at 18 months, with a consequent improvement in muscle function and cardiac hemodynamics [102].

Another recent study confirmed the long-term physiological benefits of AAV-CRISPR therapy, indicating that genome editing and dystrophin protein restoration are maintained in the *mdx* mouse model of DMD for one year after a single IV administration of AAV-CRISPR editing [103]. Moreover, this study revealed that the humoral and cellular immune response can be avoided by treating mice in the neonatal phase, contrary to the well-known immunogenic response to AAV-CRISPR therapy when injected in adult mice [104].

Taken together, preclinical AAV-CRISPR gene editing appears effective for long-term modification of pathogenic variations in the *DMD* gene with relevant therapeutic potential once efficacy, delivery, and safety issues are addressed.

The safety and efficacy of CRISPR-Cas9-based gene therapy are indeed critical points and need to be evaluated and refined before being applied therapeutically to repair mutations in human monogenic diseases. Off-target effects, delivery, and immune responses to the vectors and gene-editing system represent frequent concerns.

Particularly worrisome for long-term therapeutic use of engineered nucleases is their potential for off-target mutagenesis. Due to the extreme complexity of the genome, sgRNA may recognize other non-targeted sequences, resulting in unwanted genome-editing events; moreover, the continuous expression of Cas9 may greatly increase the odds of off-target effects. Enhancement of the length and stability of gRNA was thus developed by researchers in order to minimize off-target events [105,106].

Despite these challenges, the CRISPR/Cas9 genome-editing technology shows remarkable potential for being translated into clinical trials; one is already ongoing in treating a human genetic disease: Leber congenital amaurosis type 10 (LCA10) (NCT03872479). No clinical trials based on this approach are ongoing for DMD.

#### 4.4.2. Stem Cell Therapy

Although myofiber dystrophin deficiency is the central cause of DMD histopathology, an increasing amount of evidence suggests that DMD may also affect the function of muscle progenitor cells (MPCs) [107]. As described in the Pathophysiology of DMD section, a lack of dystrophin impacts on abnormalities in satellite cell polarity, symmetric division, and epigenetic regulation [17,108].

Restoration of dystrophin in MPCs would influence their ability to survive, self-renew, and regenerate myofibers in the dystrophic muscle.

Improvements in the characteristics of dystrophin-restored *mdx* MPCs (such as cell proliferation, differentiation, bioenergetics, and resistance to oxidative and endoplasmic reticulum (ER) stress) were observed in vitro, together with improved survival of modified MPCs upon transplantation in vivo and the ability to regenerate dystrophic muscle [107].

These studies suggested that stem cell dysfunction due to DYS deficiency is crucial for the onset and progression of muscle pathologic dystrophic features. Therefore, stem cell transplantation might be a promising method for treating DMD [109].

In the past few years, different kinds of stem cells with myogenic potential in skeletal muscle were identified, including unipotent skeletal muscle-specific stem cells, like muscle satellite cells [110], multipotent elements such as muscle-derived stem cells (MDSCs) [111], and mesangioblasts [112].

Mesoangioblasts derive from the embryonic dorsal aorta of the mouse embryo [113,114] they are considered the developmental precursors of pericytes, perivascular cells resident in the postnatal skeletal muscle [114,115]. Several studies underlined the ability of pericytes to differentiate into muscle when co-cultured with myoblasts, as well as when exposed to low-serum conditions [116]. The evidence of potential plasticity of these cells inspired their therapeutic application in DMD. In both mouse and canine models of DMD and limb-girdle muscular dystrophy, the intra-vascular injection of mouse mesoangioblasts or human pericytes showed their capacity to colonize the muscle and ameliorate clinical phenotypes [115,116,117,118]. Data resulting from the application of mesoangioblasts in preclinical models of DMD led to the first in-human, exploratory, non-randomized, open-label phase I–IIa clinical trial of intra-arterial human leukocyte antigen (HLA)-matched donor cell transplantation. Escalating doses of donor-derived mesoangioblasts were administered in limb arteries of five DMD patients under immunosuppressive therapy (tacrolimus). Four consecutive infusions were performed at 2-month intervals and, 2 months after the last infusion, a muscle biopsy was performed. In one patient, a band corresponding to the full-length dystrophin was detected by immunoblotting, but no functional improvements were reported in any of the treated patients [112].

Conversely, positive results were demonstrated by the HOPE-2 clinical trial (NCT02485938). This is a phase I/II, randomized, controlled, and open-label trial to assess the feasibility, safety, and efficacy of intra-coronary allogeneic cardiosphere-derived cells (CDCs) (named CAP-1002) in predominantly non-ambulatory DMD patients with cardiomyopathy.

Data derived from this early-phase clinical trial revealed that one single intracoronary administration of CAP-1002 did not raise serious safety concerns and provided signals of efficacy on both cardiac and upper limb function for up to 12 months. A potential systemic action of CDCs was thus supposed, suggesting that a much simpler IV delivery may be sufficient. These considerations motivated further clinical evaluation of repeated administration of IV-delivered CAP-1002 in a larger, placebo-controlled trial of DMD patients [119].

Recently, iPSCs were identified as another source for cell-based therapy of DMD. iPSCs hold great promise for the therapy of muscular dystrophy, considering their ability to rejuvenate, proliferate in vitro while keeping their pluripotency, and differentiate into multipotent cell lineages [120].

Human iPSCs (hiPSCs) derived from patients open the avenue of autologous cell therapy: transplantation of therapeutic cells differentiated from patient-derived hiPSCs will not induce immune responses observed in heterologous transplantation.

Besides the therapeutic applications, the use of patient-derived hiPSCs offered the possibility of analyzing, in vitro, the etiology and the pathophysiology of many types of muscular dystrophies, the role of genetic and epigenetic modifiers, as well as setting up in vitro protocols of gene editing before their application in vivo [121,122,123,124,125,126,127].

CRISPR/Cas9 editing was performed in DMD-iPSCs to excise exons 45–55, the most frequently deleted exons (60%) in DMD patients. Cardiomyocytes and skeletal muscle myotubes derived from reframed hiPSC clonal lines expressed stable and functional dystrophin that improved membrane stability and restored the DGC member, β-dystroglycan, in immunocompromised *mdx* mice [124].

To translate iPSCs into clinical trials, safety issues and potential limitations must be carefully addressed, together with some key aspects such as somatic cell source, optimization of delivery route, and muscle targeting. 

## 5. Conclusions and Future Directions

Several different and innovative approaches (gene based, cell based, and pharmacological) have been developed in order to restore functional dystrophin in DMD muscles. These strategies are promising, and several clinical trials are ongoing or have already been carried out on DMD patients, however, poor targeting and low efficiency in fibrotic dystrophic muscle are still hindering gene and cell-based therapies. Therefore, it is increasingly evident that future therapeutic approaches should include combined therapies, also taking into account the state of the muscle and the secondary effects induced by the lack of dystrophin.

In the DMD context, combined therapy may also be able to treat the secondary consequences of muscular dystrophy (i.e., inflammation, fibrosis, or degeneration) that decrease the efficacy of cell-based therapies and affect the accessibility of all therapeutic agents to the muscle fibers (gene, cell, or pharmacological). For these reasons, a synergic approach would have two consequences: (1) improve the muscle phenotype per se and (2) pre- or co-condition the muscle environment that is receiving the treatment in order to optimize a therapeutic response and clinical outcomes [128].

Many combinatory studies could be designed and tested: for example, it would be interesting to investigate whether ezutromid (SMT-C1100) shows a synergistic effect when combined with other therapies. Ezutromid (SMT-C1100) is an utrophin modulator, shown to be safe for healthy adult men [129]. However, a phase 2 trial with this compound (NCT02858362), sponsored by Summit Therapeutics, was terminated due to a lack of efficacy.

Other molecules belonging to the same family of ezutromid with known protective effects in *mdx* mice [130], as well as an in vitro modulator of the utrophin promoter like nabumetone [131], have been considered by Summit Therapeutics. Indeed, utrophin modulation remains a promising therapeutic strategy for all DMD patients, irrespective of their dystrophin mutation, but probably not efficient enough to be self-standing.

Over the past decades, a plethora of other experimental approaches have undergone evaluation in both *mdx* mice and clinical trials. These approaches aimed not only at restoring function to dystrophin but also at counteracting the associated processes that contribute to the disease’s progression. Anti-inflammatory factors, utrophin upregulators, compounds/drugs that regulate calcium signaling dysregulation, increased oxidative stress, mitochondrial dysfunction, accumulation of fibrosis, and defective angiogenesis constitute a wide group of promising therapeutic strategies [132,133].

Overall, DMD is experiencing many therapeutic approaches whose successes and failures will allow for the design of future effective cures.

## Figures and Tables

**Figure 1 jcm-10-00820-f001:**
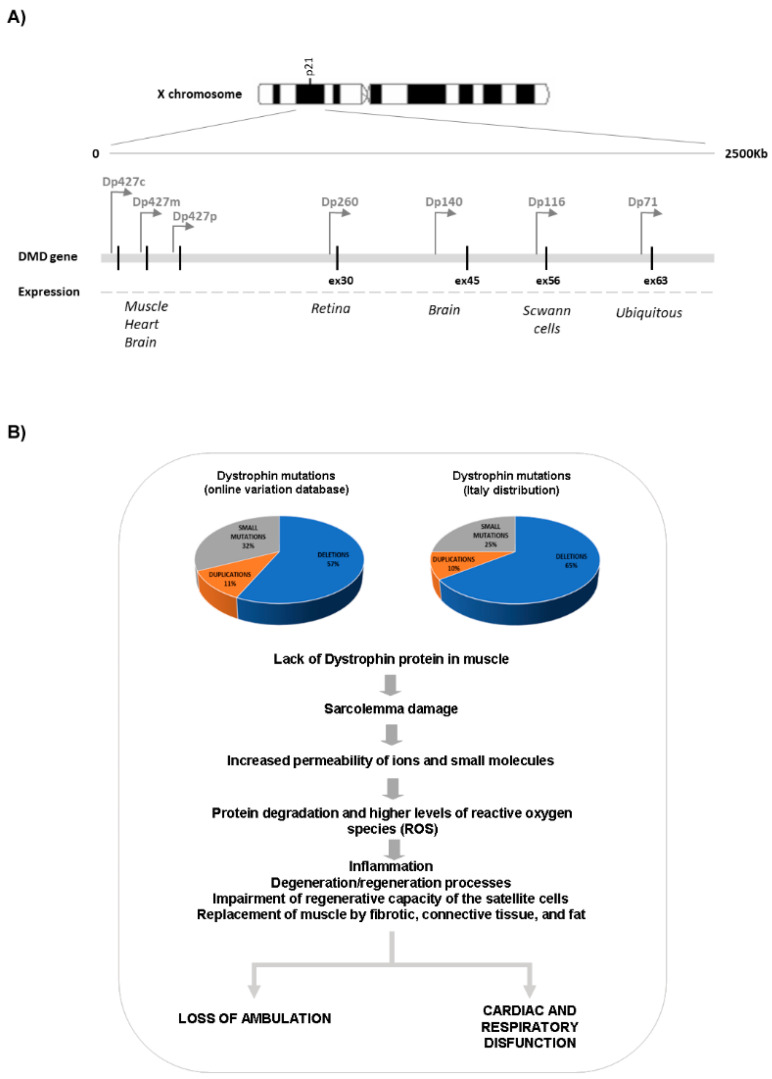
Dystrophin isoforms and effect of *DMD* mutations. (**A**) Schematic representation of dystrophin isoforms. The picture shows the localization of each promoter along the *DMD* gene (arrows) and the tissue specificity. (**B**) The two graphs report the frequency of deletions, duplications, and small mutations as described both in many reports (left graph; [9,10,11] and in a large, Italian *DMD* gene mutation study recently published by the authors (right graph, [3])). As summarized in the bottom of the panel, the *DMD* mutations lead to the absence of dystrophin protein, inducing the activation of several biological processes that cause the progressive muscle weakening and loss of ambulation, together with respiratory and cardiac complications.

**Figure 2 jcm-10-00820-f002:**
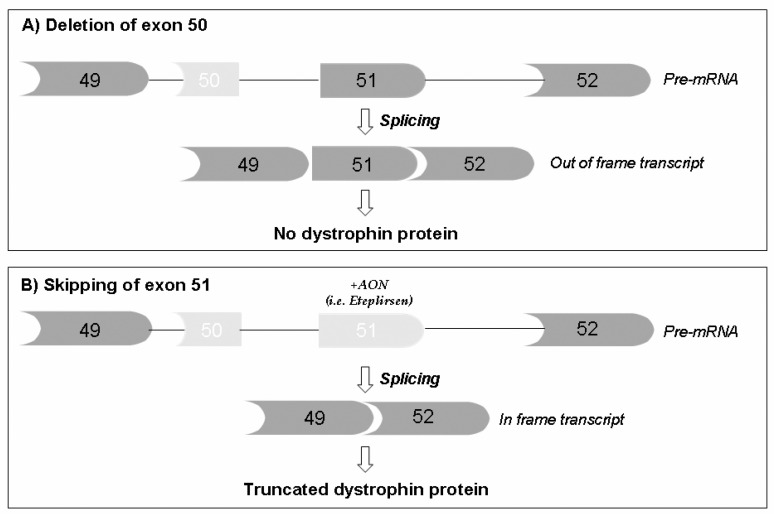
Schematic representation of exon skipping in the *DMD* transcript. (**A**) In DMD patients, the deletion of exon 50 (light gray block) generates an out-of-frame transcript that contains a premature stop codon, leading to the absence of the dystrophin protein. (**B**) The skipping of exon 51 using antisense oligonucleotides (AONs) targeting exon 51, such as eteplirsen (Exondys 51; NCT03218995, NCT03985878; NCT03992430; NCT04179409), restores the open reading frame, resulting in the synthesis of a short but functional dystrophin protein.

**Table 1 jcm-10-00820-t001:** Clinical trials (ongoing or recently terminated) evaluating exon skipping induced by AON treatments in DMD patients (http://clinicaltrials.gov (accessed on 15 January 2021)).

Chemical Modification	Therapeutic Molecule	Company	Skipped Exon	Clinical Trial	Phase	Duration
**Phosphorodi-amidate morpholino oligomers (PMO)**	Eteplirsen	Sarepta Ther.	Exon 51	NCT03218995 NCT04179409 NCT03992430 NCT03985878	Phase 2 Phase 2 Phase 3 Phase 2	2017–2021 2020–2022 2020–2026 2019–2027
	Golodirsen	Sarepta Ther.	Exon 53	NCT04179409 NCT02500381 NCT03532542	Phase 2 Phase 3 Phase 3	2020–2022 2016–2023 2018–2026
	Casimersen	Sarepta Ther.	Exon 45	NCT04179409 NCT03532542	Phase 2 Phase 3	2020–2022 2018–2026
	Viltolarsen	NS Pharma, Inc.	Exon 53	NCT03167255 NCT04060199	Phase 2 Phase 3	2017–2021 2020–2024
**2′-O-** **Methyl-phosphorothioates (2’OMePS)**	Drisapersen	BioMarin Pharmaceutical	Exon 51	NCT02636686	Phase 3	2015–2018
	DS-5141b	Daiichi Sankyo Co., Ltd.	Exon 45	NCT04433234	Phase 2	2020–2022
**Peptide phosphorodiamidate morpholino oligomer (PPMO)**	SRP-5051	Sarepta Ther.	Exon 51	NCT03675126 NCT04004065	Phase 1/2 Phase 2	2018–2024 2019–2022
**Stereopure**	Suvodirsen	Wave Life Sciences Ltd.	Exon 51	NCT03907072	Phase 2/3	2019–2020

**Table 2 jcm-10-00820-t002:** Chemical modifications of AON molecules used in DMD therapy.

Name	Chemical modification	Properties
Phosphorodiamidate morpholino oligomers (PMO)	- Substitution of the pentose sugar with a morpholine ring and the phosphate with a neutral phosphordiamidate linkage Uncharged molecules	Advantages - Resistance to nucleases, proteases, esterases, and other enzymes - Poor interaction with proteins - Low toxicity Disadvantages - Inefficient in vivo delivery Fast clearance
Peptide phosphorodiamidate morpholino oligomer (PPMO)	Conjugation with the cell-penetrating peptide (CPP)	Advantages - Enhanced tissue exposure - Greater exon skipping and dystrophin production - No negative renal or other laboratory findings in PPMO with the optimization of peptide chemistry (i.e., SRP-5051, Sarepta) Disadvantages Nephrotoxicity correlated with arginine content of the CPP
2’-O-methyl-phosphorothioates (2’OMePS)	- Incorporation of phosphorothioate (PS) linkages in non-bridging oxygen atoms and replacement of the phosphate group with sulfur - 2′-O-modification of the ribose residue Negatively charged molecules	Advantages - Nuclease resistance - High efficacy - Increased circulation - Binding to proteins in plasma and cells - Reduced renal clearance Disadvantages Toxicity and adverse effects due to retention in the kidneys and liver
Stereopure 2’OMePS	Stereochemical and chemical purity (defined stereochemistry at each PS linkage)	Advantages - Enhanced potency in cultured cells - Safe and well-tolerated Disadvantages No induction of dystrophin expression in vivo

**Table 3 jcm-10-00820-t003:** Overview of AAV-mediated gene therapies in DMD clinical trials (http://clinicaltrials.gov (accessed on 15 January 2021)).

Sponsor	Clinical Trials. Gov Identifier	Trial Name	Study Phase	Drug Name	AAV-Serotype	Primary Outcome	Secondary Outcome	Side Effects
**Solid Biosciences, LLC**	NCT03368742	Micro-dystrophin Gene Transfer Study in Adolescents and Children With DMD (IGNITE DMD)	Phase I and II, open-label, randomized, controlled	SGT-001	AAV-9 Muscle (skeletal and cardiac) tissue tropism	Safety and microdystrophin expression in biopsy	/	A serious adverse event (SAE) characterized by complement activation, thrombocytopenia, a decrease in red blood cell count, acute kidney injury, and cardio-pulmonary insufficiency
**Sarepta Therapeutics, Inc.**	NCT03375164	Systemic Gene Delivery Clinical Trial for Duchenne Muscular Dystrophy (DMD)	Phase I and II, open-label, non-randomized	rAAVrh74.MHCK7. Micro-dystrophin	AAV-rh74 Muscle (skeletal and cardiac) tissue tropism	Safety	Microdystrophin expression in biopsy and motor performances	No SAEs; Adverse events reported: elevated γ-glutamil transpeptidase; transient nausea
**Pfizer**	NCT03362502	A Study to Evaluate the Safety and Tolerability of PF-06939926 Gene Therapy in Duchenne Muscular Dystrophy	Phase Ib, open-label, non-randomized	PF-06939926	AAV-9 Muscle (skeletal and cardiac) tissue tropism	Safety and tolerability	Micro-dystrophin expression in biopsy	Three SAEs fully recovered: persistent vomiting; acute kidney injury with atypical hemolytic uremic syndrome (aHUS)-like complement activation; thrombocytopenia with aHUS-like complement activation

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
