# Peer review of "Innovative Therapeutic Approaches for Duchenne Muscular Dystrophy"

_jcm, 2021, doi:10.3390/jcm10040820_

Round 1
Reviewer 1 Report
Overall, the manuscript can be a good review of the current and future therapeutic approaches to treat DMD and this reviewer thinks it would be useful for the field after a major revision and fixing below items.
Major comments:
1- There are several sentences with long and complex structures, which makes it hard to understand (such as lines 13-15, 34-35 and so on). Therefore, using a professional English editor is needed to fix the grammatical errors throughout the script and improve readability.
2- Fig.1 image has low resolution which needs to be replaced.
3- For section 4.1 addition of a schematic figure explaining exon skipping is recommended.
4- For the same section, addition of a table summarizing pros and cons of different types of AONs is recommended.
5- In section 4.2 please add more info regarding primary endpoints of the studies (NCT00592553 and NCT01826487) and possible reason for their failure.
6- For section 4.3 addition of a table comparing different types of AAVs is recommended (target organs, efficacy, side effects, etc.).
7- Regarding AAV delivery of GALGT2, addition of more information about its function and mechanism of action to treat DMD is required.
8- Regarding AAV-CRISPR, please discuss their possible role and shortcomings in case of DMD patients with large deletions.
Author Response
- There are several sentences with long and complex structures, which makes it hard to understand (such as lines 13-15, 34-35 and so on). Therefore, using a professional English editor is needed to fix the grammatical errors throughout the script and improve readability.
The text was revised by a native English speaker.
- Fig. 1 image has low resolution which needs to be replaced.
Fig. 1 has been replaced with an image with high resolution.
- For section 4.1 addition of a schematic figure explaining exon skipping is recommended.
A new figure (Figure 2) has been added and a legend was provided.
- For the same section, addition of a table summarizing pros and cons of different types of AONs is recommended.
A new table (Table 2) has been added and a legend was provided.
- In section 4.2 please add more info regarding primary endpoints of the studies (NCT00592553 and NCT01826487) and possible reason for their failure.
The authors added more information as suggested by the reviewer.
- For section 4.3 addition of a table comparing different types of AAVs is recommended (target organs, efficacy, side effects, etc.).
A new table (Table 3) has been added and a legend has also been provided.
- Regarding AAV delivery of GALGT2, addition of more information about its function and mechanism of action to treat DMD is required.
More information about AAV delivery of GALGT2, its function, and mechanism of action to treat DMD has been added as suggested by the reviewer.
- Regarding AAV-CRISPR, please discuss their possible role and shortcomings in case of DMD patients with large deletions.
The possible role of AAV-CRISPR and its limitations in cases of DMD patients with large deletions has been added.
Reviewer 2 Report
The authors Fortunato et al., attempted to summarize innovative approaches for Duchenne muscular dystrophy in the submitted review articles ‘ innovative therapeutic approaches for Duchenne Muscular Dystrophy’. They first overlooked innate mechanism of DMD gene function as well disease pathology. They introduced some back-ground information of the disease discovery. Afterwards they lay largely their effort on introducing exon skipping, gene therapy as well cell therapy. Despite authors’ efforts in putting together a vast diverse aspects of DMD, the review article lacks coherence and some therapies were poorly described. Another major issue is that the article cited many other review articles rather than original articles. The entire article are made of very short paragraphs which is unusual and brings a lot of confusion.
- In the abstract as well as the entire article, dystrophin gene was described to be identified in 1986. However, the cell paper by Koenig et al., was published in 1987. The date of when the gene was discovered needs to be reconfirmed.
- In the abstract session they claimed first successful approved orphan drugs. So far, all the approved novel therapies for DMD has not been so successful as ataluren was not approve by FDA and Exondy 51 was only conditionally approved by FDA. They all have a lot of controversy over them. The authors need to have a balanced view as so far there is not single very successful therapy for DMD yet.
- In the introduction session, they attribute the slow progression of the DMD therapy discovery to muscle as a large organ which seems to be inappropriate.
- Also in the introduction session, the authors mentioned genetic diagnosis is mandatory which is true, however, the relevance of this part to the entire article is ambiguous and makes the review unfocused. And this is an issue to the rest of review article too.
- In the section 3 pathophysiology of DMD, the authors failed to include the cardiomyocyte which one of the major cause of the death of DMD.
- In section 4, table one does not contain all the necessary information for example, the year of the clinical trial and at which phase the clinical trial is and the name of the companies or institutions that initiated trials.
- In section 4.4.1 CRISPR/Casp9 part, the author failed to mention the most important model-dog model for the therapy and did not cite enough the most current paper that successfully restored the dystrophin in DMD dogs.
- In 4.4.2, the authors spend large paragraphs to describe Mesangiblasts which failed the clinical tests but did not describe enough other cell emerging cell therapies.
Author Response
1. In the abstract as well as the entire article, dystrophin gene was described to be identified in 1986. However, the cell paper by Koenig et al., was published in 1987. The date of when the gene was discovered needs to be reconfirmed.
The authors modified in the abstract the date of discovery of the DMD gene.
2. In the abstract session they claimed first successful approved orphan drugs. So far, all the approved novel therapies for DMD has not been so successful as ataluren was not approve by FDA and Exondy 51 was only conditionally approved by FDA. They all have a lot of controversy over them. The authors need to have a balanced view as so far there is not single very successful therapy for DMD yet.
We thank the reviewer. In the text, it was specified that Ataluren was approved only by EMA and Exondys 51 was only conditionally approved by FDA.
3. In the introduction session, they attribute the slow progression of the DMD therapy discovery to muscle as a large organ which seems to be inappropriate.
The authors thank the reviewer for the comment and deleted the sentence in the introduction session.
4. Also in the introduction session, the authors mentioned genetic diagnosis is mandatory which is true, however, the relevance of this part to the entire article is ambiguous and makes the review unfocused. And this is an issue to the rest of review article too.
The authors cannot understand this comment, sorry. Genetic diagnosis is indeed compulsory in order to get appropriate therapies and to enter into clinical trials.
5. In the section 3 pathophysiology of DMD, the authors failed to include the cardiomyocyte which one of the major cause of the death of DMD.
In section 3 mechanisms altered in DMD cardiomyocytes have been added.
6. In section 4, table one does not contain all the necessary information for example, the year of the clinical trial and at which phase the clinical trial is and the name of the companies or institutions that initiated trials.
The authors modified the table as suggested by the reviewer.
7. In section 4.4.1 CRISPR/Casp9 part, the author failed to mention the most important model-dog model for the therapy and did not cite enough the most current paper that successfully restored the dystrophin in DMD dogs.
The authors discussed the suggested paper paper as proposed by the reviewer.
8. In 4.4.2, the authors spend large paragraphs to describe Mesangioblasts which failed the clinical tests but did not describe enough other cell emerging cell therapies.
Other emerging cell therapies were added and described by the authors.
Round 2
Reviewer 1 Report
The revision covered all raised concerns and looks fine for acceptance.
Author Response
The authors thank the reviewer.
Reviewer 2 Report
The authors improved the manuscript significantly by adding in more information and correcting the tables. However, one of the concern that has not been revised is the relative short paragraphs and lack of the connecting phrases to connect the information.
Author Response
We have revised again English language, regarding review structure, we cannot see the point raised by the referee (short paragraph?), and we hope the further revision may be considered sufficient to make the article well readable.